# Influences of Different Extraction Techniques and Their Respective Parameters on the Phytochemical Profile and Biological Activities of *Xanthium spinosum* L. Extracts

**DOI:** 10.3390/plants12010096

**Published:** 2022-12-24

**Authors:** Octavia Gligor, Simona Clichici, Remus Moldovan, Dana Muntean, Ana-Maria Vlase, George Cosmin Nadăș, Gabriela Adriana Filip, Laurian Vlase, Gianina Crișan

**Affiliations:** 1Department of Pharmaceutical Botany, Iuliu Hațieganu University of Medicine and Pharmacy, 8 Victor Babeș Street, 400347 Cluj-Napoca, Romania; 2Department of Physiology, Iuliu Hațieganu University of Medicine and Pharmacy, 8 Victor Babeș Street, 400347 Cluj-Napoca, Romania; 3Department of Pharmaceutical Technology and Biopharmaceutics, University of Medicine and Pharmacy, 8 Victor Babeș Street, 400347 Cluj-Napoca, Romania; 4Department of Microbiology, Faculty of Veterinary Medicine, University of Agricultural Sciences and Veterinary Medicine, 3/5 Mănăștur Street, 400372 Cluj-Napoca, Romania

**Keywords:** *Xanthium*, innovative extraction methods, anti-inflammatory activity, antioxidant, ultrasound-assisted extraction, biological activity

## Abstract

The aim of this study was to identify possible influences of extraction methods as well as extraction parameters on the phytochemical and biological profiles of *Xanthium spinosum* L. extracts. Extraction methods were chosen as follows: classical methods, maceration and Soxhlet extraction; innovative extraction methods, turboextraction, ultrasound-assisted extraction, and a combination of the latter two. Extracts were subjected to total polyphenolic and flavonoid content spectrophotometric analysis. The phytochemical profile was determined for the best-yielding extracts using HPLC-MS analysis. Following the newly acquired data, another sorting of the extracts was performed. Biological activities such as antimicrobial and anti-inflammatory actions were evaluated, as well as oxidative stress reduction potential, on a Wistar rats inflammation model. Comparable results were achieved with Soxhlet extraction and ultrasound-assisted extraction, both surpassing all other tested methods in terms of yields. Bioactive compound concentrations tended to increase with the increase in extraction time and temperature. These maximal values lowered once the degradation points of the bioactive compounds were reached. Extracts demonstrated good protection against Gram-negative bacteria. Additionally, they provided good cellular protection and increased the antioxidant defense in the analyzed rat plantar tissue.

## 1. Introduction

The plant genus *Xanthium*, a member of the *Asteraceae* family, contains herbaceous plants with annual life cycles. *Xanthium spinosum* L., also known as spiny cocklebur, prickly burweed, etc., presents a slender stem, lined at intervals with sharp, yellow, pronged spines. The leaves are irregularly lobed and situated alternately along the stem. The fruits, called burs, are covered with small hooks. The burs are easily dispersed by attaching themselves to various surfaces. Initially found only in sandy areas such as riverbanks and coastal dunes, the genus is currently considered to have a cosmopolitan distribution, having spread to manmade environments such as cultivated lands, railway lines, and ruderal sites, due to human intervention [1,2,3].

Representatives of this genus are now considered invasive, competitive weeds in crops with great economic importance such as soybean, cotton, and peanut. Yuan et al. have reported that *X. spinosum* L. leaves release phytotoxic compounds—xanthanolides, from the class of sesquiterpene lactones, affecting neighboring plants [4]. With the possibility of being considered natural herbicides, xanthanolides also exhibit biological activities, such as anti-tumor, anti-inflammatory, antimicrobial, etc. activities—leading to species of *Xanthium* being used in traditional medicines throughout the world [3,5]. 

The bioactive compounds secreted by the members of the *Xanthium* genus have been reported to present many other potential uses such as capping and reducing agents in the synthesis of plant-derived nanoparticles, with various biological activities ranging from cytotoxicity, anti-allergic, and antimicrobial activities, and even possible applications in food waste management [6,7,8]. Furthermore, Khadom et al. have reported that *Xanthium strumarium* L. leaf extracts act as an efficient and environmentally friendly corrosion material for low-carbon steel [9]. In addition, the seed hull of the above-mentioned species was reported to provide decolorization through multilayer absorption towards Rhodamine B, one of the most used dyeing agents in several industries around the world, once again adding to the numerous potential uses of this apparently noxious plant genus [10]. 

In recent years, the characteristics of extraction processes for plant bioactive compounds that have come to be considered desirable include less time-consuming processes, fewer steps in the extraction process, usage of low-polluting and nontoxic solvents in reduced quantities, and obtaining fewer polluting or toxic by-products. Such qualities have successfully complied with newly imposed environmental regulations by authorities worldwide [11]. Extraction techniques that meet all the previously enumerated features are named innovative extraction techniques. Examples of such extraction techniques include microwave-assisted extraction, ultrasound-assisted extraction, pressurized liquid extraction, and many others. Classical or conventional extraction techniques are maceration, Soxhlet extraction, reflux extraction, and many others. These types of extraction techniques generally utilize organic volatile toxic solvents, require longer extraction times, or have numerous steps in the process. Additionally, in comparison with the innovative extraction techniques, classical techniques frequently employ harsh conditions for the plant material, such as high temperature and time values, which eventually affect the quality of the obtained extracts by degrading thermosensitive bioactive compounds [12,13,14,15].

Currently, few scientific reports have studied the influence of extraction method and parameters on *Xanthium sp*. extracts’ phytochemical and biological profiles [16,17].

Consequently, the purpose of this article was to study the influence of extraction parameters on several biological activities of *Xanthium spinosum* L. extracts obtained by different extraction methods—maceration, Soxhlet extraction, turboextraction and ultrasound-assisted extraction—while also attempting to compare the effectiveness of each method for this particular subject of interest. Based on these data, the biological effects of the selected *Xanthium spinosum* L. extract were tested on an experimental model of plantar inflammation in Wistar rats induced by carrageenan administration. The anti-inflammatory effect was quantified using oxidative stress parameters and cytokine proinflammatory secretion in the paw tissues. 

## 2. Results

In total, 20 extracts were obtained, based on the different parameters modified within the extraction processes. Thus, one sample was obtained through maceration (M), three samples were obtained through SE (S), six samples were obtained using turboextraction (T), and nine samples were obtained with UAE (U), with the last sample being obtained using UTE (UT). The samples were screened for TPC, TFC, and antioxidant capacity, according to the methods detailed above. The samples were named based on the extraction method abbreviation, followed by the parameters that were varied in each case: M for maceration, S for SE, U for ultrasound, and UT for the combination of the last two extraction methods. Table 1 provides a detailed explanation of the nomenclature of the extract samples that were evaluated in this study.

Further, the samples presenting the highest values were selected for HPLC analysis. Once the phytochemical characterization of the samples was completed, the samples containing the highest concentrations of bioactive compounds were selected for determination of biological activity. 

### 2.1. Influence of Extraction Parameters on TPC and TFC Values

The obtained values for TPC and TFC are illustrated in Table 2. Regarding the TPC values of the extracts, the highest levels of polyphenolic compounds were extracted by means of SE, with the extraction time of 60 min being the most favorable, followed by the extraction time of 20 min. However, the TPC levels of the extracts obtained using the innovative extraction methods, TBE and UAE, reached only half the values of the previous method. For TBE, the variations in time and speed showed no significant differences in results. In the case of UAE, time and temperature proved to be key factors for the yield quality, with yields increasing proportionately with the temperature and extraction time. Extreme values of the extraction time, i.e., 10 min and 30 min, lowered yields, regardless of temperature.

In relation to TFC values, UAE offered the highest yield, with 40 °C and 30 min being the optimum conditions. For UAE, increasing the temperature while maintaining a constant extraction time of 10 min decreased yields, whereas for the time value of 20 min, this resulted in increased yield levels. The second-highest value of TFC was provided by SE, with the extraction time of 60 min being once again the most favorable condition.

### 2.2. Influence of Extraction Parameters on Antioxidant Capacity

Table 3 presents values related to the antioxidant capacity. Results vary in this respect, depending on the radical and method used to measure the antioxidant capacity. Firstly, for the DPPH assay, one of the extracts achieved through TBE, i.e., having the parameters of 20 min extraction time and 4000 rpm speed, yielded the highest TE concentration. Similarly, augmenting speed while maintaining this longer time value led to an abruptly decrease in TE values for later extracts. This was also observed in the case of the lower time value for this extraction method, since TE values also decreased for the 10 min extraction proportionately with the increase in speed, although more subtly. High TE values were also achieved with SE, with the contrary difference consisting of values slowly increasing with the prolonging of extraction time. 

The FRAP and ABTS^+^ assays did not reveal significant differences between the extraction methods. The highest TE concentration measured in the FRAP assay corresponded to maceration and UAE presented the highest TE levels for the ABTS^+^ assay, namely the conditions of 10 min extraction time and 40 °C temperature. Apart from this sole example, the UAE extracts did not exhibit significant differences between the varying parameters for these last two assays.

### 2.3. HPLC-MS Analysis of the Extracts

Once the 20 extracts were screened for TPC, TFC, and antioxidant capacity, those presenting the highest values for multiple assays were selected for HPLC-MS analysis. For this step, 11 samples were analyzed. The results are presented in Table 4, Table 5 and Table 6, based on the classes of compounds investigated.

#### 2.3.1. Analysis of Polyphenolic Compounds

As shown in Table 4, three polyphenolic compounds were quantified in all of the analyzed samples: protocatechuic, vanillic, and chlorogenic acid. The samples containing these particular compounds varied depending on the extraction method, and even extraction parameters. Firstly, protocatechuic acid was found to predominate in samples obtained by maceration and SE, specifically when subjected to the 60 min extraction time. For vanillic acid, once again the SE extract with 60 min extraction time showed the most promising result, closely followed, however, by one of the UAE extracts, namely the one obtained at 20 min and 50 °C. The macerate presented only the third-highest vanillic acid concentration. Regarding the chlorogenic acid, UAE was again the most successful method to extract this particular compound, with the extraction conditions of 30 min and 40 °C resulting the highest yield. Other UAE conditions that could be considered favorable in extracting this compound are once again the 20 min extraction time, but with the temperature values of 40 °C and 50 °C. Other extraction methods with similar, albeit lower results were the SE method and the UTE. Interestingly, p-coumaric acid and caftaric acid were each successfully quantified in only two separate samples, specifically, the 60 min SE extract for the first and the UAE extract with 20 min extraction time and 40 °C for the latter.

#### 2.3.2. Analysis of Flavonoid Compounds

In total, 5 flavonoid compounds were quantified, as presented in Table 5. Isoquercitrin and quercitrin were quantified in 10 of the 11 samples subjected to analysis. The sample that did not contain quantifiable amounts of any of the studied flavonoid compounds was the macerate.

The UAE extracts exhibited the highest levels of isoquercitrin. The 30 min and 40 °C conditions achieved the highest concentration, followed by the 20 min extraction time and identical temperature. This fact would suggest a lack of significance between the two different time parameters. The same UAE conditions yielded the highest concentration of quercitrin, only in this case, they were followed by those of the 60 min SE extract. For UAE extracts, a gradual increase in concentration can be observed, proportionate to the increase in the parameters’ values. Interestingly, kaempferol was quantified only in the UAE samples and the SE extract. The other samples did not present any quantifiable amounts of this compound, regardless of the parameters. The UAE extract of 40 min and 40 °C offered once again the highest yields. However, only two other extracts managed to offer quantifiable amounts of rutin and hyperoside, namely the UAE extract of 20 min and 40 °C for rutin, and the UTE for hyperoside. 

#### 2.3.3. Analysis of Sterolic Compounds

As seen in Table 6, three sterolic compounds were quantified in the selected samples: stigmasterol, β-sitosterol, and campesterol. Some similarities were observed among the different extraction methods, with higher yields achieved with UAE and the lowest to even nonexistent with TBE. The respective UAE conditions were, once again, the extraction times of 20 and 30 min, and temperatures of 40 and 50 °C. These were closely followed by maceration and SE in the case of stigmasterol and β-sitosterol. The highest campesterol yields were obtained using SE and maceration.

### 2.4. Determination of Antimicrobial Activity

#### 2.4.1. Antimicrobial Activity—In Vitro Qualitative Study

The screening technique performed with the disk diffusion test was designed to identify the potential of the extracts to inhibit the growth of a category of microbes. An overall increased efficiency was demonstrated against Gram-negative bacteria, moderate against Gram-positive, and reduced against *Candida albicans*. The values are presented in Table 7. The diameter of the inhibition areas for Gram-positive species ranged from 6.36 to 8.94 mm; for Gram-negative strains, between 11.29 and 14.43 mm; while for *Candida albicans,* from 9.54 to 10.21 mm. The results showed an increased antimicrobial potential against Gram-negative bacteria. 

#### 2.4.2. Antimicrobial Activity—In Vitro Quantitative Evaluation

The initial antimicrobial screening revealed a good potential against Gram-negative bacteria, but we evaluated the quantitative antimicrobial potential using the MIC method against all microbial species selected for the initial qualitative assessment. The microbial potential of the extracts varied, as observed in Table 8, with lower MICs, this time, observed for Gram-positive species. 

### 2.5. Assessment of Oxidative Stress and Proinflammatory Markers

Assessment of oxidative stress was accomplished through the evaluation of the lipid peroxidation marker MDA, non-enzymatic endogenous antioxidants (reduced glutathione noted GSH, oxidated glutathione GSSG, and their ratio GSH/GSSG), as well as enzymatic antioxidants (catalase (CAT) and glutathione peroxidase (GPx)). Results are illustrated in Figure 1. 

MDA levels, evaluated in the paw tissues at 2 h and 24 h after induction of inflammation, decreased in the Indom group compared to the control group (*p* < 0.05 and *p* < 0.001). The SE extract did not significantly diminish the MDA levels in soft tissue homogenates (*p* > 0.05). It is well known that polyphenolic compounds can stimulate the activity of antioxidant enzymes, conferring protection against oxidative stress. The selected 60 min SE led to endogenous antioxidant defense increase for non-enzymatic markers (GSH, GSSG, and GSH/GSSH ratio), both at 2 h (*p* < 0.01) and at 24 h (*p* < 0.001). The 60 min SE extract provided good cellular protection and increased the antioxidant enzyme CAT and GPx activities. The maximum activity was registered at 24 h after SE administration for CAT activity (*p* < 0.001) and 2 h for GPx activity (*p* < 0.001) compared to CMC. Indomethacin decreased the GPx effect at 24 h after induction of plantar inflammation (*p* < 0.05) and did not influence the non-enzymatic antioxidant levels. 

The anti-inflammatory effect was quantified through measuring cytokine levels, IL-6 and TNF-α, in plantar tissue homogenates in comparison to indomethacin at 2 h and 24 h after carrageenan injection. The results are presented in Figure 2. Indomethacin reduced TNF-α secretion in plantar tissue at 2 h after induction of inflammation (*p* < 0.05) while SE extract decreased the TNF-α level but statistically insignificantly (*p* > 0.05). Indomethacin administered before carrageenan injection decreased IL-6 levels, particularly at 2 h, while at 24 h the values were close to those of the SE or CMC treatments (*p* > 0.05).

## 3. Discussion 

To this day, according to the scientific information gathered by the authors, few studies have reported the influence of extraction techniques and extraction parameters on the phytochemical composition of *Xanthium sp.* extracts, much less on extracts derived from the particular species *Xanthium spinosum* L. Romero et al. have optimized an aqueous extraction process of xanthatin from the aerial parts of *Xanthium spinosum* L. and studied its cytotoxic effect on a human cancer cell line [16]. Likewise, Ingawale et al. have optimized a methanolic ultrasound-assisted extraction procedure for *Xanthium strumarium* L. fruit and have studied the influence of the extraction parameters on the antioxidant, α-glucosidase inhibitory, and antimicrobial activities of the obtained extracts [17]. *Xanthium spinosum* L. extracts have demonstrated antibacterial properties and inhibitory effects on 5-lipooxygenase and cyclooxygenase-2 activities and on NFkB activation [18]. Moreover, xanthatin, a compound isolated from *Xanthium* plants, decreased nitric oxide (NO) and reactive oxygen species (ROS) generation, and reduced proinflammatory cytokines (TNF-α, IL-1β, and IL-6) levels in macrophages RAW 264.7 pretreated with lipopolysaccharide (LPS). Additionally, the expression of STAT3, ERK1/2, SAPK/JNK, IκBα, and p65 was improved after xanthatin pretreatment, suggesting a beneficial effect on inflammatory conditions [19].

Our findings regarding the TPC and TFC are in accordance with Kumar et al. and Ingwale et al., as initial increase in extraction time leads to higher yields by enhancing the cavitation effect; however, long exposure to ultrasound induces damage to the plant material, lowering extraction yields [17,20]. Similarly, high extraction temperatures led to polyphenol compound degradation [21]. Thus, an extraction time of 20 min and a temperature of 40 °C resulted in the highest yields for the UAE extracts. Additionally, regarding the FRAP results of the samples, previous reports stated that only solvent concentrations and solid-to-solvent ratios influence FRAP results for UAE extracts of *X. strumarium* L. fruits [17]. Flavonoid compounds such as rutin and hyperoside were noted to be selectively present in samples of UAE, U24, and UT. A possible explanation for this finding might reside in the degradation of the compounds at elevated temperatures and prolonged extraction time. Additionally, the lack of these compounds in extracts obtained using milder conditions might be due to the insufficiency of the extraction processes. A similar explanation could be applicable to the lack of the sterolic compound campesterol in sample T48, though it was present in the majority of the samples. The corresponding parameters in this case were 8000 rpm rotation speed and 20 min extraction time (four cycles of 10 min), both maximum values for TBE, possibly leading to the degradation of the sterolic compound [22].

Although an important protective response of the organism against harmful agents such as microorganisms or cell and tissue damage, inflammation may become difficult to manage in chronic or acute cases, as well as negatively impacting the subject’s safety and quality of life. Despite the plethora of available anti-inflammatory drugs that decrease or even prevent inflammation in the body, scientists, medical professionals, and patients alike are still faced with adverse reactions resulting from such treatments. This troublesome aspect opens potential therapeutic windows for bioactive compounds as alternative forms of treatment, with the advantages of lack of adverse reactions, biological safety, and proven efficacy [23,24].

Similar to our findings, a potential against Gram-negative bacteria was also observed by Ghahari et al. in fruit essential oil of the species *X. strumarium* L. [25]. Scherer et al. reported no differences between antimicrobial potential against *S. aureus*, *E. coli*, and *P. aeruginosa* strands for hydroalcoholic SE leaf extracts of *X. strumarium* L.; however, the extracts exhibited strong overall antimicrobial activity [26]. Similar activity against *S. aureus* was reported by Ingawale et al. for methanolic *X. strumarium* L. fruit extracts [17].

Methanolic as well as aqueous extracts of several plant parts from different *Xanthium* species, in different animal models, were also reported to lower oxidative stress and proinflammatory marker levels. As such, a possible mechanism for the anti-inflammatory effect is related to the inhibitory action on NF-kB, STAT1, and MAPK activation and consequently to the reduced secretion of proinflammatory cytokines [3,27,28,29,30].

## 4. Materials and Methods

### 4.1. Plant Material

Commercially available *Xanthium spinosum* L. dried aerial material was purchased from a local tea company (Hypericum Impex, Baia Sprie, Maramureș, Romania). The plant material was ground to a coarse powder with a Bosch MKM6003 grinder (Gerlingen, Germany), according to the European Pharmacopoeia, 10th edition. The plant material presented a moisture content of 10%.

### 4.2. Chemicals and Reagents

Aluminum chloride (AlCl3), Folin-Ciocâlteu reagent, indomethacin, carboxymethylcellulose, o-phthalaldehyde, Lambda carrageenan type IV, sodium carbonate (Na_2_CO_3_), ABTS (diammonium 2,2′-azino-bis(3-ethylbenzothiazoline-6-sulfonate), DPPH (2,2-Diphenyl-1-(2,4,6-trinitrophenyl) hydrazyl), and TPTZ (2,4,6-Tris(2-pyridyl)-s-triazine), were purchased from Sigma–Aldrich (Taufkirchen, Germany). 2-thiobarbituric acid and Bradford reagent were obtained from Merck KGaA (Darmstadt, Germany) and ELISA tests for cytokines (TNF-α, IL-6) were purchased from Elabscience (Houston, TX, USA). The Bradford total protein assay was obtained from Biorad (Hercules, CA, USA). All HPLC reagents and standards were of analytical grade and were acquired from Sigma–Aldrich (Germany) and Decorias (Rediu, Romania), respectively.

### 4.3. Extraction Processes

Throughout all extraction procedures, the solvent-to-sample ratio was kept at a constant 1:10 (*w/v*), in order to better uniformize results and to provide clearer conditions for the comparison of the extraction methods. The chosen solvent was 70% ethanol.

#### 4.3.1. Maceration

The extraction was performed according to the indications of the Romanian Pharmacopoeia. In a Falcon flask, 50 mL 70% alcohol was added to 5 g plant material and left for 10 days at room temperature, with periodical agitation. Separation was realized through centrifugation at 12000 rpm for 10 min.

#### 4.3.2. Soxhlet extraction (SE)

As stated above, the same proportions were maintained for this extraction. A SER 148 solvent extraction unit (VELP^®^ Scientifica, Usmate Velate, Italy) was utilized. The temperature was kept constant throughout the process, with time being the studied parameter. The selected values for extraction time were 20, 40, and 60 min. Separation was further realized through centrifugation at 12000 rpm for 10 min.

#### 4.3.3. Turboextraction (TBE)

This extraction process was carried out by means of a T 50 ULTRA-TURRAX^®^ disperser (IKA^®^-Werke GmbH & Co. KG, Staufen, Germany). The studied parameters were extraction time and speed. The plant material was added to a flask along with the solvent and subjected to dispersion for two cycles of 5 min and 4 cycles of 5 min. The extraction time was chosen to be discontinued in order to prevent device overheating and solvent evaporation. The selected speed values were 4000, 6000, and 8000 rpm. Once the extraction process was concluded, separation was further realized through centrifugation at 12000 rpm for 10 min.

#### 4.3.4. Ultrasound-Assisted Extraction (UAE)

The ultrasound-assisted extraction was performed using a Sonic-3 ultrasonic bath (Polisonic, Warsaw, Poland), with frequency and power kept constant at 50 Hz and 230 V, respectively. Time and temperature were the studied parameters in this case. Therefore, the selected values were 30°, 40°, and 50 °C for temperature and 10, 20, and 30 min for extraction time. After extraction, samples were subjected to separation by centrifugation at 12000 rpm for 10 min.

#### 4.3.5. Combination of UAE and TBE (UTE)

The selected temperature for the ultrasonic bath was 30 °C and for the ULTRA-TURRAX^®^ disperser, the selected speed value was 4000 rpm. The chosen extraction time was one cycle of 5 min. These parameters were maintained constant and selected as such in order to prevent overheating of the disperser and potential evaporation of the solvent. The obtained sample was then subjected to separation by centrifugation at 12000 rpm for 10 min.

### 4.4. Determination of Total Phenolic Content (TPC)

The total polyphenolic content (TPC) was quantified using the Folin-Ciocâlteu method implemented by Csepregi et al. with slight modifications in terms of volumes and reaction time [31]. Thus, 270 µL Folin-Ciocâlteu reagent was added to 60 µL plant extract, followed by the addition of 270 µL Na_2_CO_3_ 6% (*w/v*) in microtubes. After 30 min incubation in the dark, the absorbances were measured at 765 nm against a gallic acid standard. Results were expressed as mg gallic acid equivalents per ml extract (GAE mg/mL).

### 4.5. Determination of Total Flavonoid Content (TFC)

The total flavonoid content (TFC) was determined using the method of Pinacho et al. with subsequent modifications [32]. A 200 µL volume of plant extract as mixed with 400 µL solution consisting of AlCl_3_ 20 mg/mL in 5% acetic acid in ethanol in a 3:1 (*v/v*) ratio. The absorbance was measured at 420 nm with quercetin as a standard. Results were expressed as mM quercetin equivalents (QE mM).

### 4.6. Antioxidant Activity Analysis

#### 4.6.1. DPPH Radical Scavenging Activity

The DPPH assay was performed by adapting the protocol provided by Martins et al. with respective modifications [33]. A 200 µL volume of extract was mixed with 800 µL DPPH radical methanolic solution and left for incubation for 30 min at a temperature of 40 °C away from light. Absorbances were measured at 517 nm, with Trolox reagent serving as standard. Results were expressed as mg Trolox equivalents per ml extract (TE mg/mL). 

#### 4.6.2. ABTS^+^ Scavenging Activity

The ABTS^+^ assay was carried out based on the method offered by Erel [34]. A 200 µL volume of acetate buffer (0.4 M, pH 5.8) was added to 20 µL ABTS^+^ in acetate buffer (30 mM, pH 3.6). Further, 12.5 µL extract was added to the previous mix. Absorbances were measured at 660 nm, with Trolox as a standard. Results were expressed as mM Trolox equivalents (mM TE).

#### 4.6.3. FRAP Assay

The FRAP assay was accomplished following the method used by Csepregi et al., with slight modifications. A 30 µL volume of extract was mixed with freshly prepared FRAP reagent [31]. The reagent was obtained by adding together 25 mL acetate buffer (300 mM, pH 3.6), 2.5 mL TPTZ solution (10 mM TPTZ in 40 mM HCl), and 2.5 mL FeCl_3_ (20 mM in water). After incubation at room temperature for 30 min, the absorbances were measured at 620 nm. Trolox was used as a standard, with results given in mM Trolox equivalents (TE mM).

### 4.7. HPLC-MS Analysis

The identification and quantification of bioactive compounds from plant extracts was performed using several validated liquid chromatography tandem mass spectrometry analytical methods. The used apparatus was an Agilent 1100 HPLC Series system (Agilent Technologies, Santa Clara, CA, USA) equipped with binary pump, degasser, column thermostat, UV detector, and autosampler. This system was coupled with a mass spectrometer, type Brucker Ion Trap SL (Brucker Daltonics GmbH, Leipzig, Germany). For the separation of 29 different polyphenols, a reverse-phase analytical column was employed (Zorbax SB-C18, 100 × 3.0 μm i.d., 3.5 μm particle size) and two distinct analytical methods were applied. 

The first analytical method was used to identify the following polyphenols: caftaric acid, gentisic acid, caffeic acid, caffeoylquinic acid, chlorogenic acid, p-coumaric acid, ferulic acid, sinapic acid, vitexin, hyperoside, vitexin 2-O-rhamnoside, isoquercitrin, rutoside, myricetin, fisetin, quercitrin, kaempferitrin, quercitol, kaempferol-3-rhamnoside, patuletin, luteolin, kaempferol, and apigenin. Both UV and MS mode were used for compound detection. For the UV detector the wavelength was set at 330 nm until 17.5 min of analysis and then was changed to 370 nm until the end of analysis. The ionization source of the MS system was an electrospray operating in negative mode. For polyphenol carboxylic acids, the MS operated in monitoring specific ions mode, while for flavonoids and their aglycones the AUTO MS mode was selected. Separation was performed using a mobile phase of methanol:acetic acid 0.1% (*v/v*) and binary gradient elution which started with a linear gradient (from 5% to 42% methanol at 35 min), then was kept at isocratic elution for the following 3 min (with 42% methanol). Afterwards, the column was rebalanced with 3% methanol. The injection volume was of 5 μL and the flow rate was set at 1 mL/min [35,36,37].

A second LC-MS analytical method was employed to identify the following polyphenols: epicatechin, catechin, syringic acid, gallic acid, protocatechuic acid, and vanillic acid. The same column as described previously was selected for chromatographic separation. The mobile phase consisted of methanol:acetic acid 0.1% (*v/v*). A binary gradient was used for elution, as follows: at start—3% methanol; at 3 min—8% methanol; from 8.5 to 10 min—20% methanol; for rebalancing the column—3% methanol. The injection volume was 5 μL and the flow rate 1 mL/min. For detection of the polyphenolic compounds, the MS mode (SIM-MS) was selected. The MS system operated under the same conditions as aforementioned [38,39].

For sterol compounds identification, the following analytical standards were used: ergosterol, beta-sitosterol, campesterol, and stigmasterol. Separation was performed with the same analytical column as for polyphenol compounds and under isocratic elution conditions. The mobile phase consisted of methanol:acetonitrile 10:90 (*v/v*). The positive ion mode monitoring was selected for MS analyses and was realized with the Agilent Ion Trap 1100 SL MS apparatus with an APCI interface. For identification of sterol compounds, the MS spectra and RTs were compared with those obtained under the same conditions for standard compounds. To decrease the background interference, the multiple reaction monitoring analysis mode (MS/MS) was selected [40,41]. 

For chromatographic data acquisition and analysis, the Data Analysis (v5.3) and ChemStation (vA09.03) software from Agilent Inc. (Santa Clara, CA, USA) were used. 

### 4.8. Determination of Antimicrobial Activity

#### 4.8.1. Antimicrobial activity—In Vitro Qualitative Study

The evaluation of the antimicrobial potential was performed in two steps. Initially, the disk diffusion test as a screening method was used to identify the extracts with high antimicrobial potential against standard strains of Gram-positive and Gram-negative bacteria and yeasts. The microbial strains selected for the study were represented by four Gram-positive: *Staphylococcus aureus* ATCC 6538P, *Listeria monocytogenes* ATCC 13932, *Enterococcus faecalis* ATCC 29212, and *Bacillus cereus* ATCC 11778; three Gram-negative: *Escherichia coli* ATCC 10536, *Salmonella enteritidis* ATCC 13076, and *Pseudomonas aeruginosa* ATCC 27853; and one yeast strain: *Candida albicans* ATCC 10231. Amoxicillin for bacteria and ketoconazole for yeasts were used as standard antibacterial and antifungal controls.

Screening was performed with EUCAST standards, using an adapted disk diffusion method [42]. Young microbial colonies (24 h old) previously grown on Mueller–Hinton (MH) agar for bacteria and Sabouraud dextrose agar (SDA) were used to prepare a suspension adjusted at 0.5 density in saline on a McFarland scale using a Densichek calibration standard (bioMérieux, France). The suspension was used to flood 8.5 cm diameter plastic Petri dishes with MH agar for bacteria and SDA agar for yeast. After removing the excess fluid, the agar surface was allowed to dry, and 5 mm diameter filter paper discs were placed in a radial model. A total amount of 20 µL was placed on each filter paper disk and the plates [42]. Antimicrobial activity was assessed by measuring the diameter of the growth inhibition area, expressed in mm.

#### 4.8.2. Antimicrobial Activity—In Vitro Quantitative Evaluation

The quantitative assessment included the minimum inhibitory concentration (MIC) method for the same eight standard microbial strains. This evaluation was performed according to the modified EUCAST protocols [42]. The method was performed using 96-wells titer plates containing the extracts diluted in liquid MH medium inoculated with 20 µL from the microbial suspension. The stock solutions of the extracts were diluted using a two-fold serial dilution system in ten consecutive wells, from the initial concentration (1/1) to the highest (1/512). The total broth volume was adjusted to 200 µL. Positive controls represented by microbial inoculum in MH broth and negative control represented by microbial inoculum in 30% ethanol were also prepared and used to fill wells 11 and 12, respectively. The plates were incubated for 24 h at 37 °C for bacteria and 48 h at 28 °C for *Candida*. MIC values were determined as the lowest concentration of the extracts’ dilution that inhibited the growth of the microbial cultures (having the same OD as the negative control), compared to the positive control, as established by a decreased value of absorbance at 450 nm (HiPo MPP-96, Biosan, Latvia). The MIC50 was also determined, representing the MIC value at which ≥50% of the bacterial/yeast cells were inhibited in their growth, considered as the well with the OD value similar to the average between the positive and negative control.

### 4.9. Evaluation of Biological Activities

Following the phytochemical profiling of the samples, the 60 min SE was selected for the additional assessment of in vivo biological activities. For selection, the extract presenting the largest range of identified compounds and with the highest yields, particularly those of high interest for this study, were taken into consideration.

#### 4.9.1. Carrageenan-Induced Inflammation Model in Rats

An in vivo study was carried out using an experimental model of paw inflammation in male Wistar rats (110–130 g). Following a week of acclimatization in the following conditions: 12 h light/12 h dark cycles, 35% humidity, free access to water, and a normocaloric standard diet (VRF1); the animals were randomly divided into 4 groups of 8 specimens each. Oral gavage treatment was administered once a day for 4 days, in a maximum volume of 0.25 mL, as follows: group 1—carboxymethylcellulose 2% (positive control group—CMC); group 2—Indomethacin 5 mg/body weight (b.w.) in carboxymethylcellulose 1.5% (Indom); group 3 15 mg TPC/b.w./day (60 min SE). 

On the fifth day, inflammation induction was performed through injection in the right hind footpad of 100 µL of freshly prepared 1% carrageenan (λ-carrageenan, type IV, Sigma) diluted in normal saline [43]. The exact volume of saline solution was injected in the left hind footpad, bearing the role of negative control. Afterwards, under anesthesia with 90 mg/kg ketamine and 10 mg/kg xylazine, samples of soft paw tissues were collected at 2 and 24 h after carrageenan administration. These samples were used for oxidative stress parameters and cytokine level assessment. For oxidative stress and inflammation evaluation, the samples of soft tissues were homogenized using a Brinkman Polytron homogenizer (Kinematica AG, Littau-Luzern, Switzerland) in 50 mMTRIS–10 mM EDTA buffer (pH 7.4) as previously published [43]. The protein content was measured with the Bradford method [44]. 

Experimental procedures were approved by the Ethic Committee Board of “Iuliu Hațieganu” University of Medicine and Pharmacy, Cluj-Napoca, Romania (291/23.02.2022) on animal welfare according to the Directive 2010/63/EU on the protection of animals used for scientific purposes.

#### 4.9.2. Oxidative Stress Assessment

In order to evaluate oxidative stress, the malondialdehyde (MDA), glutathione reduced and glutathione oxidized levels, and GSH/GSSG ratio were measured in paw tissue homogenates. MDA levels were quantified with spectrofluorimetry, using the 2-thiobarbituric acid method, while GSH and GSSG levels were determined using the Hu method [45,46].

#### 4.9.3. Proinflammatory Cytokine Evaluation

The concentration of TNF-α and IL-6 in plantar tissue homogenates was evaluated with an ELISA assay according to the manufacturer’s protocol. Results were expressed as pg/mg protein.

#### 4.9.4. Statistical Analysis

The data were analyzed with a one-way analysis of variance (ANOVA) followed by the Tukey’s multiple comparisons post-test using GraphPad Prism 8 software. A *p* value < 0.05 was considered statistically significant. The results were expressed as mean ± standard deviation. 

## 5. Conclusions

Total content analysis as well as the phytochemical profile of the different *Xanthium spinosum* L. extracts were determined. It could be concluded that both most representative extraction methods of the two categories, i.e., SE of the classical extraction methods and UAE of the innovative methods, obtained comparable results, with similar variations based on extraction parameters. However, SE represented overall better yield levels. Biologically, the SE sample demonstrated good cellular protection and increased the antioxidant enzyme activity, as well as antimicrobial activity. For this reason, one could conclude that *Xanthium spinosum* L., spiny cocklebur or prickly burweed, presents an area of scientific interest for complementary therapeutical areas, such as nutraceuticals, herbal food supplements, or adjuvant therapies in the pharmacological treatment of acute or chronical inflammatory diseases.

## Figures and Tables

**Figure 1 plants-12-00096-f001:**
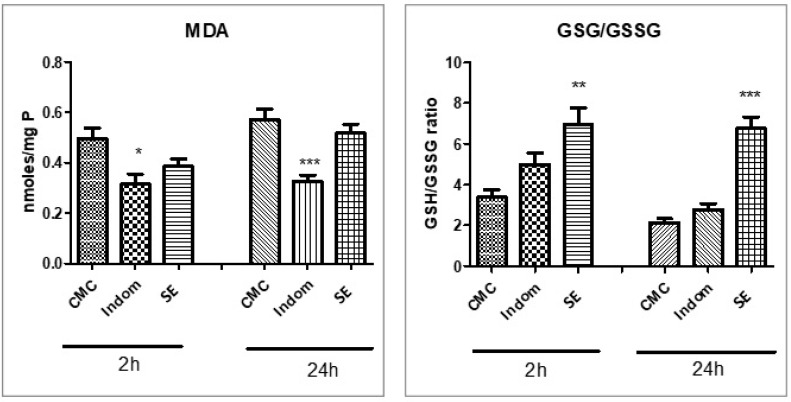
MDA levels, GSG/GSSG ratio, CAT, and GPx activities in soft paw tissue at 2 and 24 h after carrageenan injection in rats pretreated with XS extract. Values are means ± SD. Statistical analysis was performed using a one-way ANOVA, with Tukey’s multiple comparisons post-test (* *p* < 0.05, ** *p* < 0.01 and *** *p* < 0.001 vs. control group).

**Figure 2 plants-12-00096-f002:**
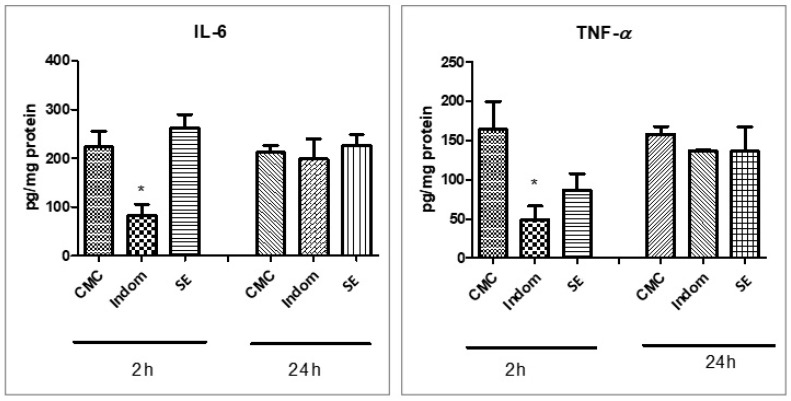
IL-6 and TNF-α levels in paw tissue of the experimental rats at 2 and 24 h after carrageenan injection. Values are means ± SD. Statistical analysis was performed using a one-way ANOVA, with Tukey’s multiple comparisons post-test (* *p* < 0.05 vs. control group).

**Table 1 plants-12-00096-t001:** Nomenclature of the evaluated extract samples.

Extraction Method	Studied Extraction Parameters				Sample Name
Maceration	*				M
Soxhlet extraction (SE)	Time (min)	20			S20
		40			S40
		60			S60
Turboextraction (TBE)		10 min (2 cycles of 5 min)	Rotation speed (rpm)	4000 6000	T24
					T26
				8000	T28
		20 min (4 cycles of 5 min)		4000	T44
				6000	T46
				8000	T48
Ultrasound-assisted extraction (UAE)		10	Temperature (°C)	30	U13
				40	U14
				50	U15
		20		30	U23
				40	U24
				50	U25
		30		30	U33
				40	U34
				50	U35
Combination of UAE and TBE (UTE)	**				UT

* Parameters remained constant, see Section 4.3.1. Maceration, ** Parameters remained constant, see Section 4.3.5. Combination of UAE and TBE (UTE).

**Table 2 plants-12-00096-t002:** TPC and TFC of the extracts.

Sample	TPC (GAE mg/mL) *	TFC (QE mM) *
M	0.296 ± 0.016	0.526 ± 0.033
S20	0.533 ± 0.003	0.406 ± 0.014
S40	0.440 ± 0.010	0.304 ± 0.022
S60	0.564 ± 0.003	0.597 ± 0.004
T24	0.316 ± 0.022	0.195 ± 0.006
T26	0.344 ± 0.006	0.214 ± 0.000
T28	0.280 ± 0.013	0.077 ± 0.025
T44	0.306 ± 0.009	0.136 ± 0.004
T46	0.306 ± 0.004	0.227 ± 0.014
T48	0.153 ± 0.002	0.070 ± 0.055
U13	0.067 ± 0.002	0.298 ± 0.012
U14	0.104 ± 0.004	0.205 ± 0.009
U15	0.191 ± 0.002	0.176 ± 0.004
U23	0.147 ± 0.007	0.300 ± 0.010
U24	0.372 ± 0.006	0.455 ± 0.028
U25	0.254 ± 0.004	0.488 ± 0.007
U33	0.141 ± 0.009	0.332 ± 0.010
U34	0.279 ± 0.010	0.697 ± 0.013
U35	0.331 ± 0.011	0.217 ± 0.010
UT	0.248 ± 0.012	0.478 ± 0.035

* Concentrations are expressed as mean ± SD.

**Table 3 plants-12-00096-t003:** Antioxidant capacity of the extracts.

Sample	DPPH (TE mg/mL) *	FRAP (TE mM) *	ABTS^+^ (TE mM) *
M	0.923 ± 0.284	8.741 ± 0.119	1.732 ± 0.191
S20	1.317 ± 0.184	3.540 ± 0.118	2.641 ± 0.191
S40	1.379 ± 0.128	3.213 ± 0.052	1.985 ± 0.262
S60	1.518 ± 0.066	4.611 ± 0.005	2.843 ± 0.431
T24	1.058 ± 0.411	3.428 ± 0.027	1.152 ± 0.473
T26	1.035 ± 0.144	3.760 ± 0.152	1.581 ± 0.191
T28	0.939 ± 0.052	3.966 ± 0.082	1.354 ± 0.493
T44	1.653 ± 0.032	4.488 ± 0.171	1.581 ± 0.306
T46	1.013 ± 0.135	5.484 ± 0.095	1.581 ± 0.374
T48	0.536 ± 0.204	5.974 ± 0.239	0.823 ± 0.287
U13	0.729 ± 0.029	3.871 ± 0.000	1.354 ± 0.191
U14	0.643 ± 0.008	4.424 ± 0.027	6.732 ± 0.342
U15	1.310 ± 0.031	4.867 ± 0.000	1.833 ± 0.152
U23	0.698 ± 0.017	3.008 ± 0.109	1.960 ± 0.087
U24	1.423 ± 0.008	4.915 ± 0.126	2.465 ± 0.158
U25	1.124 ± 0.022	7.587 ± 0.137	4.207 ± 0.558
U33	0.709 ± 0.029	3.885 ± 0.071	2.111 ± 0.116
U34	1.371 ± 0.025	4.820 ± 0.082	2.717 ± 0.087
U35	0.925 ± 0.037	4.361 ± 0.072	2.742 ± 0.000
UT	0.964 ± 0.008	4.361 ± 0.119	2.439 ± 0.076

* Concentrations are expressed as mean ± SD.

**Table 4 plants-12-00096-t004:** Polyphenolic compounds in the selected extracts.

Sample	Protocatechuic Acid (µg/mL Extract) *	Vanillic Acid (µg/mL Extract) *	Chlorogenic Acid (µg/mL Extract) *	*p*-Coumaric Acid (µg/mL Extract) *	Caftaric Acid (µg/mL Extract) *
M	0.31 ± 0.037	0.25 ± 0.752	13.54 ± 0.542	<LOQ	<LOQ
S60	0.29 ± 0.023	0.31 ± 0.028	28.49 ± 1.424	1.58 ± 0.111	<LOQ
T26	0.08 ± 0.007	0.20 ± 0.016	16.33 ± 0.490	<LOQ	<LOQ
T44	0.11 ± 0.014	0.14 ± 0.020	16.56 ± 0.993	<LOQ	<LOQ
T46	0.11 ± 0.011	0.18 ± 0.021	14.14 ± 1.131	<LOQ	<LOQ
T48	0.04 ± 0.005	0.10 ± 0.004	9.01 ± 0.631	<LOQ	<LOQ
U14	0.08 ± 0.010	0.18 ± 0.021	23.50 ± 3.525	<LOQ	<LOQ
U24	0.10 ± 0.014	0.19 ± 0.027	28.33 ± 0.850	<LOQ	1.82 ± 0.055
U25	0.07 ± 0.004	0.30 ± 0.039	30.68 ± 4.601	<LOQ	<LOQ
U34	0.08 ± 0.009	0.18 ± 0.013	37.47 ± 2.623	<LOQ	<LOQ
UT	0.06 ± 0.009	0.21 ± 0.030	31.81 ± 3.817	<LOQ	<LOQ

* Concentrations are expressed as mean ± SD; <LOQ below limit of quantification.

**Table 5 plants-12-00096-t005:** Flavonoid compounds in the selected extracts.

Sample	Kaempferol (µg/mL Extract) *	Isoquercitrin (µg/mL Extract) *	Quercitrin (µg/mL Extract) *	Rutin (µg/mL Extract) *	Hyperoside (µg/mL Extract) *
M	<LOQ	<LOQ	<LOQ	<LOQ	<LOQ
S60	0.35 ± 0.011	2.05 ± 0.061	17.19 ± 1.547	<LOQ	<LOQ
T26	<LOQ	1.58 ± 0.047	7.47 ± 0.672	<LOQ	<LOQ
T44	<LOQ	1.28 ± 0.128	7.66 ± 1.072	<LOQ	<LOQ
T46	<LOQ	0.97 ± 0.087	7.66 ± 0.230	<LOQ	<LOQ
T48	<LOQ	0.66 ± 0.066	4.85 ± 0.146	<LOQ	<LOQ
U14	0.28 ± 0.014	2.35 ± 0.330	10.09 ± 1.513	<LOQ	<LOQ
U24	0.28 ± 0.020	3.59 ± 0.323	12.33 ± 0.740	15.55 ± 1.711	<LOQ
U25	0.42 ± 0.012	3.28 ± 0.492	15.14 ± 0.454	<LOQ	<LOQ
U34	0.55 ± 0.082	4.05 ± 0.445	20.56 ± 0.617	<LOQ	<LOQ
UT	0.42 ± 0.058	3.28 ± 0.361	15.14 ± 0.605	<LOQ	1.86 ± 0.167

* Concentrations are expressed as mean ± SD; <LOQ below limit of quantification.

**Table 6 plants-12-00096-t006:** Sterolic compounds in the selected extracts.

Sample	Stigmasterol (ng/mL Extract) *	β-Sitosterol (ng/mL Extract) *	Campesterol (ng/mL Extract) *
M	6116.07 ± 733.929	38,089.10 ± 4570.692	616.59 ± 18.498
S60	5467.96 ± 820.193	32,740.34 ± 3928.840	663.46 ± 39.808
T26	2926.73 ± 263.406	10,905.65 ± 327.170	425.56 ± 29.790
T44	4544.47 ± 181.779	20,648.84 ± 1651.907	438.09 ± 52.571
T46	3525.19 ± 423.023	18,160.25 ± 2179.230	291.00 ± 37.830
T48	567.38 ± 62.412	1947.76 ± 194.776	<LOQ
U14	5248.96 ± 157.469	25,350.01 ± 2535.001	440.67 ± 61.693
U24	5874.92 ± 469.994	27,462.53 ± 3844.754	475.70 ± 57.084
U25	7920.66 ± 792.066	38,442.64 ± 3075.411	521.30 ± 36.491
U34	8296.93 ± 580.785	42,135.04 ± 5477.555	533.12 ± 69.305
UT	6816.47 ± 681.647	35,124.95 ± 2458.747	454.43 ± 27.266

* Concentrations are expressed as mean ± SD; <LOQ below limit of quantification.

**Table 7 plants-12-00096-t007:** The results of the qualitative screening technique—the disk diffusion test.

	U34	S60	Amoxicillin	Ketoconazole
*Staphylococcus aureus* ATCC 6538P	6.7	8.26	24.38	-
*Enterococcus faecalis* ATCC 29212	6.67	7.79	16.8	-
*Listeria monocytogenes* ATCC 13932	6.36	6.39	18.96	-
*Bacillus cereus* ATCC 11778	7.82	8.94	8.83	-
*E. coli* ATCC 10536	14.43	13.04	13.72	-
*Salmonella enteritidis* ATCC 13076	13.47	11.29	18.43	-
*Pseudomonas aeruginosa* ATCC 27853	12.53	11.77	R	-
*Candida albicans* 10231	9.54	10.21	-	23.74

Note: Inhibition area diameter in mm; R—resistant.

**Table 8 plants-12-00096-t008:** The results of the quantitative technique—MIC test.

	U34		S60		Amoxicilin		Ketoconazole	
	MIC 100	MIC 50	MIC 100	MIC 50	MIC 100	MIC 50	MIC 100	MIC 50
*Staphylococcus aureus* ATCC 6538P	1/16	1/32	1/32	1/32	4 *	2 *	-	
*Enterococcus faecalis* ATCC 29212	1/16	1/32	1/16	1/16	8 *	4 *	-	
*Listeria monocytogenes* ATCC 13932	1/16	1/32	1/32	1/32	1 *	0.5 *	-	
*Bacillus cereus* ATCC 11778	1/16	1/32	1/32	1/64	16 *	8 *	-	
*E. coli* ATCC 10536	1/16	1/32	1/16	1/32	8 *	4 *	-	
*Salmonella enteritidis* ATCC 13076	1/8	1/16	1/8	1/16	4 *	4 *	-	
*Pseudomonas aeruginosa* ATCC 27853	1/16	1/32	1/32	1/32	R *	R *	-	
*Candida albicans* 10231	1/32	1/32	1/32	1/64	-		8 *	4 *

* Values represent the concentration in µg/mL; R—resistant.

## Data Availability

Not applicable.

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
