# Peer review of "Influences of Different Extraction Techniques and Their Respective Parameters on the Phytochemical Profile and Biological Activities of *Xanthium spinosum* L. Extracts"

_plants, 2022, doi:10.3390/plants12010096_

Round 1

Reviewer 1 Report

The comments are as follows:

1.       Present more key-findings in the abstract.

2.       The abbreviations should be defined just the first time they are mentioned in the text and used as such throughout the text.

3.       Please, provide more details regarding the plant material (moisture content, particle size).

4.       Please, use the exact name of alcohol used as a solvent for the extractions.

5.       Please, improve the discussion and comparison of the results from the sections 2.1., 2.2 and 2.3. with the literature data. Overall, the Discussion section has to be improved significantly.

6.       The importance of this study and the practical applications of the findings has to be expressed more.

Author Response

Influences of different extraction techniques and their respective parameters on the phytochemical profile and biological activities of Xanthium spinosum L. extracts

Octavia Gligor, Simona Clichici, Remus Moldovan, Dana Muntean, Ana-Maria Vlase, George Cosmin Nadăș, Gabriela Adriana Filip, Laurian Vlase, Gianina Crișan

Esteemed Editor and Reviewers,

The authors of this paper wish to express their gratitude for your feedback. Please find below the authors’ comments to the current review report.

Reviewer #1 – Comments and replies

Point

Reviewer’s comments

Authors’ comments

1

Present more key findings in the abstract.

The Abstract was expanded.

2

The abbreviations should be defined just the first time they are mentioned in the text and used as such throughout the text.

The authors would like to propose a new table, the current Table 1, in hopes of aiding the readers in understanding the nomenclature of the samples.

3

Please, provide more details regarding the plant material (moisture content, particle size).

Details were provided.

4

Please, use the exact name of alcohol used as a solvent for the extractions.

Type of alcohol was added.

5

Please, improve the discussion and comparison of the results from the sections 2.1., 2.2 and 2.3. with the literature data. Overall, the Discussion section has to be improved significantly.

Discussions section was improved.

6

The importance of this study and the practical applications of the findings has to be expressed more.

Details were added.

Reviewer 2 Report

Dear authors

Kindly give more details/explanations about the following questions:

1-    Abstract: Specific aims was studying the influences of different extraction techniques and their respective parameters on the phytochemical profile and biological activities of Xanthium spinosum L. extracts: kindly added one or two sentences regarding the obtained results in the abstract regarding this goal.

2-    Keywords: about 19 words is too much, kindly reduce the number of keywords.

3-    Introduction: generally, at the end of the introduction we announce the objective of the work, so kindly move the paragraph from line 91-98 to discussion section or in on other part of the introduction section.

4-    Results: 2.4.2.Antimicrobial activity – in vitro quantitative evaluation: move the lines 285-290 to the discussion section.

5-    Line 106-108: added the full name of the abbreviations: SE, UAE, UTE, TFC, TPC…, The samples were named based on the extraction method abbreviation, followed by the parameters which were varied in each case: its not clear and its confused, for example for U13, U14, U15…….what is the meaning of the number after U? Some time you put SE and in the table you put S60? Kindly homogenize the names of the samples and theirs related abbreviations.

6-    Table 1 and 2: correct first column on the left

7-    Table 3: Caftaric acid was detected only in U2 extract, kindly give explanation  

8-    Table 4: In the extracts U14, U25 and U34 no Rutin was detected, but in U24 extract 15.55 µg/ml extract? Kindly give explanation.

9-    Table 4: For Hyperoside it was detected only on UT extract, kindly give explanation

10-                        Table 5: Campesterol is not detected only on T48 extract, give explanation

11-                        Table 7: provide the results for the positives controls (Amoxicillin, Ketoconazole)

12-                        Kindly provide the chromatograms of the selected extracts in result section

13-                        Please provide a table with the analytical parameters of your “internal database”, such as RT, precursor m/z, and fragments.

14-                        Discussion: 19 lines for the discussion is too short and not enough for the reader to understand the obtained results. The weakest point is the poor discussion of the obtained results with previous works. The discussion section should be improved.

Author Response

Influences of different extraction techniques and their respective parameters on the phytochemical profile and biological activities of Xanthium spinosum L. extracts

Octavia Gligor, Simona Clichici, Remus Moldovan, Dana Muntean, Ana-Maria Vlase, George Cosmin Nadăș, Gabriela Adriana Filip, Laurian Vlase, Gianina Crișan

Esteemed Editor and Reviewers,

The authors of this paper wish to express their gratitude for your feedback. Please find below the authors’ comments to the current review report.

Reviewer #2 – Comments and replies

Point

Reviewer’s comments

Authors’ comments

1

Abstract: Specific aims was studying the influences of different extraction techniques and their respective parameters on the phytochemical profile and biological activities of Xanthium spinosum L. extracts: kindly added one or two sentences regarding the obtained results in the abstract regarding this goal.

Further comments were added in the Abstract regarding the study goal.

2

Keywords: about 19 words is too much, kindly reduce the number of keywords.

Keywords have been modified.

3

Introduction: generally, at the end of the introduction we announce the objective of the work, so kindly move the paragraph from line 91-98 to discussion section or in on other part of the introduction section.

Paragraph was removed and included in the Discussions section.

4

Results: 2.4.2.Antimicrobial activity – in vitro quantitative evaluation: move the lines 285-290 to the discussion section.

Lines were removed, then added to the Discussions section.

5

Line 106-108: added the full name of the abbreviations: SE, UAE, UTE, TFC, TPC…, The samples were named based on the extraction method abbreviation, followed by the parameters which were varied in each case: its not clear and its confused, for example for U13, U14, U15…….what is the meaning of the number after U? Some time you put SE and in the table you put S60? Kindly homogenize the names of the samples and theirs related abbreviations.

The authors would like to propose a new table, the current Table 1, in hopes of aiding the readers in understanding the nomenclature of the samples.

6

Table 1 and 2: correct first column on the left

Please refer to the above-mentioned answer, Point 5.

7

Table 3: Caftaric acid was detected only in U2 extract, kindly give explanation 

The quantitative levels of each compound found in the analysed extract was provided only if:

1.         The mass spectrum of the compound matches the mass spectrum of the standard (qualitative positive identification)

2.         The concentration of the compound is not below quantification limit of the analytical method.

Otherwise, if the adequate accuracy of the compound level cannot be provided (e.g. very low concentration), the compound is declared <LOQ (below limit of quantification).

Thus, to be more accurate on the provided results, in Table 4, 5…6 all 0.00 values were replaced by <LOQ.

8

Table 4: In the extracts U14, U25 and U34 no Rutin was detected, but in U24 extract 15.55 µg/ml extract? Kindly give explanation.

Same as above, <LOQ replaced the 0.00 values in the table.

9

Table 4: For Hyperoside it was detected only on UT extract, kindly give explanation

Same as above, <LOQ replaced the 0.00 values in the table.

10

Table 5: Campesterol is not detected only on T48 extract, give explanation

Campesterol level fell below quantification limit of the analytical method, <LOQ was indicated in the table.

11

Table 7: provide the results for the positives controls (Amoxicillin, Ketoconazole)

The requested data were added.

12

Kindly provide the chromatograms of the selected extracts in result section

Chromatograms were provided in the Supplementary Material. The authors would prefer to include this information in a separate, Supplementary Material, considering that these data have been already presented in previously published scientific articles, which were cited accordingly in the manuscript, and thus no longer present novelty.

13

Please provide a table with the analytical parameters of your “internal database”, such as RT, precursor m/z, and fragments.

Table was provided in the Supplementary Material. The authors would prefer to include this information in a separate, Supplementary Material, considering that these data have been already presented in previously published scientific articles, which were cited accordingly in the manuscript, and thus no longer present novelty.

14

Discussion: 19 lines for the discussion is too short and not enough for the reader to understand the obtained results. The weakest point is the poor discussion of the obtained results with previous works. The discussion section should be improved.

Discussions section was improved.

Round 2

Reviewer 1 Report

The authors responded to all reviewer's remarks.

Reviewer 2 Report

Thanks for your answers